# Flavor Profile of Tomatoes Across Different Cultivation Times Based on GC × GC-Q/TOFMS

**DOI:** 10.3390/foods14172975

**Published:** 2025-08-26

**Authors:** Yuan Gao, Nan Jiang, Jing Liu, Guanglu Cui, Meng Zhao, Yuanfang Du, Hua Ping, Cheng Li

**Affiliations:** 1Institute of Quality Standard and Testing Technology, Beijing Academy of Agriculture and Forestry Sciences, Beijing 100097, China; gaoyuan060117@163.com (Y.G.);; 2Risk Assessment Laboratory for Agro-Products (Beijing), Ministry of Agriculture and Rural Affairs, Beijing 100097, China; 3Daxing District Planting Technology Promotion Station, Beijing 102600, China; cuiguanglu@163.com

**Keywords:** tomato, GC × GC-Q/TOFMS, volatiles, cultivation time, metabolic pathway

## Abstract

Volatile compounds greatly affect tomato aroma, but systematic analysis of volatiles in tomatoes is limited by detection techniques. Here, HS-SPME Arrow-GC × GC-Q/TOFMS was employed to analyze tomato flavor profiles across different cultivation times. To investigate the effects of light and temperature on aroma profiles, three tomato samples across different cultivation periods, including S1 (harvested on May 30th, with lowest temperature and light conditions), S2 (harvested on August 10th, with the highest temperature and light), and S3 (harvested on June 27th, with moderate temperature and light), were analyzed. Overall, 227 volatiles were identified, belonging to 9 aroma categories. Hexanal, *(E)*-2-hexenal, nonanal, *(E)*-2-Octenal, *trans*-geranylacetone, 6-methyl-5-hepten-2-one, 3,4-Octadiene, 7-methyl-, and citral were found to be the key volatiles contributing most significantly to differentiating the samples across cultivation periods, imparting grassy and floral–fruity notes, respectively. The S1 tomatoes had a distinct grassy aroma, whereas the S3 tomatoes had a floral/fruity fragrance. Most differential metabolites were correlated with fatty acid, amino acid, and isoprenoid pathways. S1 tomatoes were characterized by fatty aldehydes (mainly C6/C9), and S2 tomatoes contained high concentrations of fatty alcohols. S3 tomatoes were positively correlated with isoprenoid-derived volatiles. These variations might be caused by the fluctuations in daily temperature and light intensity. This work establishes a foundational reference for assessing environmental effects on tomato flavor profiles.

## 1. Introduction

Tomato (*Solanum lycopersicum* L.) fruits are nutritious, with a unique flavor that is highly favored by consumers. Because they are rich in carbohydrates, organic acids, carotenoids, flavonoids, minerals, and various other nutrients, tomatoes are popular worldwide. Volatile aromatic compounds (volatiles), sugars, and acids in tomatoes are the source of their distinctive flavor. The flavor of tomatoes greatly affects consumer sensory preferences and acceptance of tomato products [1]. Tieman et al. [2] combined sensory evaluation panels with chemical and genomic analyses of nearly 400 varieties of tomatoes, 33 chemicals were identified that were correlated with consumer’ preferences, including fructose, glucose, citric acid, malic acid, and 29 volatile compounds [2]. This research shows that the flavor profile of tomatoes is affected by both biochemical components and genetic influences [2]. Notably, volatiles are critical in the perceived taste quality of tomatoes, and the absence of certain carotenoid-derived volatile compounds negatively affects consumer preferences [3]. There have been substantial advances in identification of the pathways controlling formation of crucial flavor compounds, such as sugars and acids. However, research on the volatile compounds that determine tomato aroma remains comparatively limited. The evolution of GC-MS techniques has played a pivotal role in this field. Nevertheless, it exhibits specific limitations in the identification of small molecular species and trace components, specifically when differentiating trace concentrations of volatile compounds within complex matrices [4]. As an enhanced analytical approach, GC × GC-Q/TOFMS demonstrates superior capability for detection of volatiles compared with conventional methods. This method has been applied to dairy products and other foods [5]. Yin et al. [6] analyzed the flavor profile of Harbin red sausage using GC × GC-qMS, and found that many volatile compounds significantly overlapped in column I due to their similar carbon numbers, and these compounds could be re-injected into column II by the modulator for better separation. Both GC/MS and GC × GC-TOFMS have been utilized to determine the volatile compounds in cherry tomatoes, with results demonstrating that GC × GC-TOFMS detected 363 more volatiles than GC/MS [7]. GC × GC-TOFMS provides distinct advantages including enhanced sensitivity, effective separation, and higher scanning rate, etc. These advantages effectively address the limitations of GC-MS.

Tomato flavor is affected by the cultivar, cultivation practices, harvest maturity and environmental factors. Although the influence of genetic factors cannot be ignored, in specific cases involving fixed varieties, environmental factors have a greater effect than phenotype when it comes to flavor. Among the environmental factors, light exposure and temperature are critical in influencing tomato quality. Research has shown that flavonoids and carotenoids concentrations in tomatoes and peppers are affected by light conditions during plant growth, and insufficient light exposure leads to decreases in the concentrations of these compounds [8]. According to another study, abaxial leaf supplemental lighting improves flavor quality in cherry tomatoes [9]. However, a large reduction in terpenoid content has been observed in grape berries with increased light exposure, and this has been primarily attributed to an increase in the fruit temperature [10]. These results indicate that temperature plays a key role in flavor development. Compared to lower temperature (9 °C), terpenes levels in carrots exhibited a positive correlation with increasing cultivation temperature (21 °C) [11]. Conversely, some terpenes levels were negatively correlated with high temperature in grape varieties [12]. Some research has suggested that the inconsistent influence of temperature on volatiles is likely because it is challenging to distinguish between the effects of temperature and radiation in field-based studies [13]. Furthermore, high temperature and exposure to full sunlight can promote carotenoid degradation [14,15], thereby generating more norisoprenoid compounds. The activity of carotenoid cleavage dioxygenase (CCD) 4, a key enzyme in norisoprenoid synthesis, is inhibited by high temperatures (>37 °C) [16]. This implies that excessively high temperatures can down-regulate gene expression, consequently affecting metabolite levels. Moreover, light and temperature exert divergent regulatory effects on volatile compounds derived from the three primary metabolic pathways [17]. Therefore, investigating the synergistic influence of light and temperature on the biosynthesis of characteristic volatile compounds is essential.

Significant variations in metabolite profiles occur across different growing conditions among plants. Nevertheless, the metabolism of volatiles in tomatoes under varying planting times remains poorly understood. In particular, the flux and partitioning of assimilated carbon into critical flavor and aroma metabolism in tomatoes across cultivation periods are still not well characterized. Variations in tomato flavor throughout the growing season are largely attributed to distinct meteorological variables, such as solar irradiance levels and temperature regimes. To address the above issues, HS-SPME Arrow extraction followed by GC × GC-Q\TOFMS was employed to characterize volatile compounds in ripening tomato fruits. The key sensory flavor attributes were elucidated by constructing a flavor wheel for the tomatoes. Furthermore, the effect of cultivation timing on tomato flavor profiles was investigated to examine the regulatory roles of light and temperature within core metabolic pathways. This work provides a foundation for further investigating the biosynthesis and regulatory mechanisms of aroma substance in tomato fruits.

## 2. Materials and Methods

### 2.1. Experimental Design

Field experiment was performed at a commercial tomato planting farm located in Daxing District, Beijing, China (39°39′48″ N, 116°33′54″ E), during March to August of 2023 growing season. Tomatoes of the variety ‘Fan Zhi Wei No. 3’ served as the experimental material in the study. For all of the three tomato samples, three biological replicates were created to sample the collection. The cultivation times of the three tomato samples were: S1, transplanted in March and harvested in late May (May 30th); S2, planted from May to August, harvested on August 10th; S3, planted from April to June, harvested on June 27th. All experimental units were subjected to the same cultivation practices, for disease prevention, nutrient supplementation, and plant structure management.

To better understand variations in the microclimate around the tomatoes across the three distinct harvest periods, data loggers and acquisition equipment were deployed near fruit clusters (Figure 1D,E). The meteorological parameters, including temperature, solar radiation, and photosynthetically active radiation (PAR), were automatically recorded every 10 min using a HOBO data logger (Onset, Bourne, MA, USA). Figure 1A–C illustrates the typical daily fluctuations of environmental factors near the harvest date. Analysis of the meteorological data revealed that S2 tomatoes experienced the highest temperature, solar radiation, and PAR, whereas S1 tomatoes experienced the lowest temperature, solar radiation, and PAR.

### 2.2. Samples Collection

Tomato fruits were harvested at the fully ripe stage for all the three samples. For each replicate, 30 fruits of the same maturity, size, and with no mechanical damage for each sampling point were collected randomly from 15 plants of each variety, placed in foam box incorporating dry ice cooling agents, and transported to the laboratory within 2 h. Sample collection was consistently conducted between 8:00 AM and 10:00 AM. The samples were immediately frozen in liquid nitrogen and stored at −80 °C until analysis. After the sample collection was complete, all samples were uniformly analyzed by GC × GC-Q\TOFMS.

### 2.3. Extraction of Volatile Compounds

When all sampling was completed, the fruits were ground to powder under liquid nitrogen. Following a modification of an established method [7], volatile compounds were extracted and analyzed. Briefly, 0.5 g of sample powder, 0.1 g of NaCl, and 1 μL of internal standard (IS, 4-methyl-2-pentanol, 100 mg/L, purity ≥ 99.9%, GC standard, Aladdin, Shanghai, China) were placed into a 20 mL headspace vial. The vial was sealed immediately and submitted to SPME. The analysis was performed using a CombiPAL3 autosampler (CTC Analytics, Zwingen, Switzerland) equipped with robotic tool change (RTC) and coupled to an Agilent 8890 GC system with a 7250 accurate-mass Q-TOF detector (Agilent Technologies Inc., Santa Clara, CA, USA). Samples were pre-incubated at 50 °C for 20 min and extracted using a Smart SPME Arrow with a 1.1 mm divinylbenzene/carboxen WR/polydimethylsiloxane (DVB/CWR/PDMS) phase (Supelco, Bellefonte, PA, USA) for 30 min at 50 °C, and then subjected to desorption for 5 min. The total GC cycle time, including oven cooling, was 85 min.

### 2.4. GC × GC-Q/TOFMS Conditions

A DB-HeavyWAX polar capillary column (60 m × 0.25 mm, 0.25 µm; J&W Scientific, Folsom, CA, USA) and a mid-polar DB-17 ms column (1.85 m × 0.18 mm, 0.18 µm; J&W Scientific) were used as the first- and second-dimension GC columns, respectively. Volatile compound modulation was achieved using an SSM 1810 solid-state modulator (J & X Technologies Co., Ltd., Shanghai, China) with a modulation period of 6.0 s, which was installed between the two columns. For separation, the temperature was initially 40 °C (1 min), increased to 250 °C at 4 °C/min, and then held at 250 °C for 10 min. The temperature of the second-dimension column was maintained at 5 °C above that of the first-dimension column. The mass spectrometry conditions were as follows. GC conditions: Inlet 250 °C; transfer line to MS 260 °C; carrier gas: high-purity helium at 1.2 mL/min with a split ratio of 5:1. MS settings: electron ionization at 70 eV, ion source 200 °C; continuous scanning from 33 to 550 *m*/*z* at 50 spectra/s.

### 2.5. Identification of Volatile Components

Retention time alignment, matched filtration, peak detection, and peak matching were performed on a Canvas 2.5 workstation (J & X Technologies Co., Ltd., Shanghai, China). The peaks were identified with SNR > 10 and qualitatively identified by matching against the NIST 20 database (similarity > 750). Subsequently, the actual linear retention indices (LRIs) were compared with theoretical LRIs. When the difference was ≤30, it was initially inferred that the two compounds matched. The actual LRI of each volatile compound was calculated using a series of n-alkanes (C7–C30), and used for compound identification in conjunction with mass spectral fragmentation data. The retention times, LRIs, CAS numbers, molecular formulas, molecular masses, and percentages of the compounds are provided in Appendix A. Volatile quantification was performed using a semi-quantitative method. The relative concentrations of tomato volatile were determined quantitatively using an IS and calculated using the equation: *C*_volatiles_ = Area_volatiles_ × *C*_IS_/Area_IS_.

### 2.6. Statistical Analyses

Data are presented as the mean ± standard deviation of at least three biological replicates for each attribute. Statistical analysis was performed with SPSS 22.0 software package (SPSS Inc., Chicago, IL, USA) for one-way analysis of variance (ANOVA) followed by Duncan’s test (*p* < 0.05). Charts were plotted using Sigma Plot 14.0 and Excel. The dataset was visualized using the pheatmap package in R (v 4.4.3) to generate hierarchical clustering heatmaps of relative metabolite quantitation values. Principal component analysis (PCA) was performed using MetaboAnalyst 6.0 (https://www.metaboanalyst.ca/, accessed on 7 June 2025).

## 3. Results and Discussion

### 3.1. Volatile Components in Tomatoes

Comprehensive two-dimensional and three-dimensional separation diagrams from GC × GC-Q/TOFMS were used to analyze tomato volatiles (Figure 2). A total of 986 peaks were detected, resulting in the identification of 227 volatiles. This is more than the number of compounds determined in previous studies using conventional GC/MS [18,19]. Compounds with similar carbon numbers overlapped in column I, and these compounds were re-injected into the second column with different polarities using the modulator to achieve better separation. For instance, 1-Butanol, 3-methyl- (I) and 2-Hexenal (II) had almost the same RT, but were separated by the second column. Similarly, 5,9-Undecadien-2-one,6,10-dimethyl- (trans-geranylacetone) (III) and Phenol, 2-methoxy- (IV) were separated on the second-dimension column but not the first-dimension column. These compounds were major components and likely had substantial contributions to tomato aroma.

### 3.2. The Identification of Crucial Volatiles in Tomatoes

HS-SPME Arrow-GC × GC-Q/TOFMS successfully identified 227 volatile compounds in the tomatoes. Among the compounds identified, hexanal and 2-hexenal-(E) had the highest relative concentration. These compounds are known for their fresh grass and green leaf aromas. The aroma-active compounds detected in the present study were classified into the following nine categories: grassy, fruity, floral, fatty, woody, nut-like, caramel, sour, and others (compounds with unclear odor characteristics) (Figure 3A). The odor perceptions of aroma-active compounds were obtained from http://www.thegoodscentscompany.com (accessed on 20 May 2025). The results suggested that during the phenological maturity phase of tomatoes, green or grassy aromas occurred most frequently (42 times), followed by woody (or herbal), fruity, fatty, and floral flavors (41, 35, 34, and 22 times). The sour, caramel, and almond or nut-like flavors were observed 9, 7, 4 times, respectively. Additionally, 33 substances exhibited ambiguous aromatic profiles that could not be clearly classified through sensory description.

A flavor wheel for key aromatic compounds with high relative concentrations was created in Figure 3B. These high concentration components were identified as characteristic compounds representing the odor classes. Specifically, the grassy aroma was primarily characterized by C6/C9 aldehydes, with hexanal and (*E*)-2-hexenal identified as the dominant representative compounds. Volatile compounds associated with caramel, almond/nut-like, and sour aromas also exhibited relatively straightforward compositions, with predominant compounds of furan derivatives, benzaldehyde compounds, and medium- chain/short-chain fatty acids, respectively. By contrast, the other four aroma categories exhibited more diverse volatile compositions. Notably, the fruity aroma was attributed to ketones (including terpene ketones), and esters. The woody and herbal flavor was linked to aldehydes, alcohols, terpenes, monoterpenoid alcohols, and sesquiterpenes. While the fatty flavor was caused by fatty alcohols and aldehydes. Finally, the floral flavor was attributed to alcohols, aldehydes, and terpenes.

### 3.3. Variations in Volatiles Composition Across Different Cultivation Times

To systematically compare the organoleptic flavor profiles among tomatoes cultivated for varying durations, a bar chart was constructed of the cumulative concentrations of distinct aroma types (Figure 4A). A sensory flavor profile radar chart (Figure 4B) was constructed using the cumulative contribution values derived from semi-quantitative concentration data. The results revealed that the green-grassy odorants had the highest content, with S1 showing the most intense grassy character among the three cultivation periods. Fatty flavors constituted the secondary category with higher levels, though no statistically significant variations were observed across these samples (*p* < 0.05). These were followed by fruity, floral and woody/herbal flavor, with S3 exhibiting the significantly highest fruity and woody flavor, while S1 showed the lowest floral flavor. The lowest woody flavor was observed in S2. These were the prominent aroma features for the volatiles detected in tomatoes in this study. No statistically significant differences were observed for the remaining three odor categories across cultivation timepoints.

The biosynthesis of volatile compounds in tomatoes is influenced by environmental factors such as the temperature, humidity, light intensity, and cultivation practices during fruit development [12,20]. Variations in dry matter content, sugar concentrations, and phenolic compounds have been observed in fruits harvested from plants grown in different seasons within the same year [21,22]. Meanwhile, significant differences in carotenoids and lycopene levels have been found in tomato fruits across different growing seasons [23]. These variations may be caused by seasonal fluctuations in the daily temperature and solar radiation intensity [23]. Previous studies have suggested that temperature and light may play a critical role in the formation of characteristic odor compounds [8,24]. Low temperatures are more conducive to the accumulation of grassy scents, whereas high temperatures promote the cleavage of carotenoid, and lead to enhanced synthesis of isoprene derivatives and more pronounced floral/fruity characteristics. However, excessively high temperatures can also negatively affect the retention of volatile flavor compounds.

### 3.4. Effect of Cultivation Time on Volatiles According to the Pathways

According to precursor-mediated biosynthetic routes, the volatile compounds were classified as fatty acid-derived volatiles (e.g., C6/C9 aldehydes and lactones), amino acid-derived volatiles (e.g., branched-chain aldehydes and alcohols) and isoprenoid-derived volatiles (terpenes and their derivatives) [25]. Some compounds with undetermined metabolic pathways were excluded from this analysis (Figure 5).

#### 3.4.1. Fatty Acid-Derived Volatiles

In tomato fruits, fatty acid-derived volatiles (FADVs), which were the most abundant among the three metabolic pathways, are predominantly synthesized from polyunsaturated fatty acids (linoleic/α- linoleic acid) through the LOX pathway [26,27], which generates fatty alcohols, aldehydes, acids, and esters. Our analysis identified 25 aldehydes, 22 alcohols, 11 ketones, 9 acids, 7 esters, and 6 alkenes. Aldehydes were dominant, with a contribution of >80% to the total fatty acid-derived volatiles concentration. This result is consistent with a previous report [28]. Among the compounds detected, C6/C9 aldehydes are characteristic aroma compounds that impart green aromas reminiscent of freshly cut grass or leaves and enhance perceived tomato freshness [29]. Key volatiles such as hexanal and (*E*)-2-hexenal had high concentrations and were critical contributors to tomato flavor, which is in agreement with previous reports [30,31]. The concentrations of these compounds in the S1 samples were significantly higher than in the S2 and S3 samples. Although certain aldehydes reached highest concentrations in S3 or S2 samples, S1 contained the greatest total aldehyde content primarily due to the dominance of the two aforementioned aldehydes (Hexanal and (E)-2-hexenal), which collectively account for approximately 40% of the total aldehyde pool. The lowest aldehyde levels occurred in the S2 samples, which experienced the highest solar radiation, the photosynthetically active radiation and temperature. These findings suggested that the aldehyde accumulation was negatively correlated with high temperatures and strong light. Yoo et al. [8] compared the effects of shaded versus non-shaded growth conditions on tomatoes and peppers, and found that fruits from the shaded plants had higher contents of green/leaf odor volatiles. These results suggested that the synthesis of these volatiles was upregulated under light deficient conditions. Previous studies have shown that C6/C9 concentrations also show light-dependent regulation, and their concentrations are reduced with light exposure [17] and elevated under canopy shading because of microclimate changes [32,33]. These results are consistent with our observations. PAR can also affect C6/C9 accumulation and shows strong negative correlations [14]. Similarly, grapes grown in cool climates typically contain high concentrations of C6/C9 compounds [12], which further supports the accumulation of C6/C9 compounds at low temperatures and is in agreement with our results.

Flavor profiles vary with the presence of different volatile functional groups. Alcohols typically contribute to the fusel, fermented, fruity, and ethereal flavor in tomatoes, and significantly enhance the overall flavor. It was observed that most of the fatty alcohols had a relatively high concentration in the S2 samples. The most abundant fatty alcohols in the ripe tomatoes were 1-hexanol, 3-hexen-1-ol, (Z)-, 2-hexen-1-ol, (Z)-, 1-pentanol, 1-octen-3-ol, 2-octen-1-ol, (E)-, 1-octanol, and 1-nonanol. These compounds were present at higher concentrations in the S2 samples than in the S1 and S3 samples, which suggested that strong light and high-temperature conditions promoted alcohol accumulation. This observation is consistent with previous research that demonstrated that leaf removal significantly increased the concentrations of C6 alcohols, particularly (Z)-3-hexenol, without affecting total soluble solids [10]. In addition to aldehydes and alcohols, ketones and esters are also present in measurable quantities and are important contributors to fruity and sweet aromas [34]. These compounds potentially influence consumer preferences. In the present study, these compounds had relatively high concentrations in the S3 samples. Compared with the S1 and S3 samples, the S2 samples contained the highest concentrations of acids. Overall, the effects of light and temperature on volatiles accumulation in the tomatoes were complex. These results indicate that there is potential for improving tomato aroma quality by varying light and temperature conditions to modulate the concentrations of volatiles.

#### 3.4.2. Amino Acid-Derived Volatiles

Volatile compounds derived from amino acid metabolism in tomatoes are primarily classified into two structural categories: benzenoids (with aromatic rings) and branched-chain aliphatic derivatives. Benzenoids are synthesized from phenylalanine, whereas branched-chain aliphatic compounds usually originate from the three branched-chain amino acids valine, isoleucine, and leucine [35]. In the present study, the total concentration of branched-chain aliphatic compounds showed almost no difference among the three samples, although some variations were observed in individual substances. Previous reports have suggested that branched-chain aliphatic compounds are unaffected by total solar radiation [35,36]. However, benzenoids had higher concentrations in the S2 samples, followed by the S3 samples, which indicated positive correlations with temperature and light. These compounds, particularly phenylethyl alcohol, contribute to floral (e.g., rose) aromas. The biosynthesis of 2-phenylacetaldehyde is regulated by aromatic amino acid decarboxylases (AADCs). Research has shown that the expression of *LeAADC1A* and *LeAADC1B* is down regulated under low temperature (4 °C) storage [37], and this leads to a decrease in 2-phenylacetaldehyde production [38]. Furthermore, a reduction in UV radiation decreases the formation of phenylalanine-derived volatile compounds and flavonoids [35,39]. Meanwhile, leaf removal can increase the concentrations of amino acid-derived volatiles in grapes by increasing light exposure [40]. In our study, the low solar/UV radiation levels in the S1 samples appeared to contribute significantly to its low benzenoid volatile levels. However, the metabolic and regulatory mechanisms of this pathway require further investigation.

#### 3.4.3. Isoprenoid-Derived Volatiles

Many volatiles in tomatoes are derived from isoprenoid metabolism, which predominantly yields terpenoids and norisoprenoids. These compounds are synthesized from C5 isoprene precursors through the MEP pathway (Figure 5). 62 terpenoids and 14 norisoprenoids were identified in the tomatoes in this study. The S3 samples contained higher levels of isoprenoid-derived volatiles than the S1 and S2 samples.

Linalool, trans-nerolidol, (Z)-Vertocitral C, and 3,4-Octadiene, 7-methyl- were the primary volatile terpenes in the tomatoes. The highest concentrations of these volatiles were found in the S3 samples, which suggested that adequate light exposure and an appropriate temperature were required to promote terpene biosynthesis. The terpene biosynthesis pathway begins with C5 isoprene unit production, and this is followed by oxidation and glycosylation [41]. Consequently, factors affecting the activities of key enzymes and transcriptional regulation of biosynthetic genes may drive changes in the characteristic volatiles in tomatoes [42]. This may contribute to structural diversification and metabolic complexity of tomato terpenoids. Both light and temperature regulate terpenoid biosynthesis. For instance, light exposure promotes terpene accumulation in *Cabernet Sauvignon* grapes [17]. Studies also suggest that light supplementation directly enhances cyclic monoterpene biosynthesis in Japanese mint [43]. These findings show that light and temperature might be the two important environmental factors regulating fruit flavor. Fu et al. [20] investigated light–temperature interactions regulating volatiles in strawberries postharvest. They found that the conditions that most affect terpene concentrations were temperature and darkness. In a study in carrots, terpene concentrations increased with higher growth temperatures (18 and 21 °C) compared with lower temperatures (9, 12, and 21 °C) [11]. However, contrasting results were observed in *Vitis* cultivars, where most terpenes (γ-terpinene, terpinen-4-ol, *cis*-furan linalool oxide, and *trans*-pyran linalool oxide) exhibited negative correlations with increasing temperature [12]. In carrots, terpinolene concentrations decreased with increases in the growth temperature [11]. The results from these temperature studies are inconsistent, which is likely because of the challenge in separating temperature and radiation effects in field trials [13].

Regarding norisoprenoids, they are predominantly generated from the cleavage of carotenoids catalyzed by carotenoid cleavage dioxygenases (CCDs). As the predominant carotenoids in tomato fruits, lycopene and β-carotene serve as primary precursors. Integrated metabolomic and proteomic analyses have revealed that the synthesis of key norisoprenoids such as trans-geranylacetone, citral, and β-ionone are originated from lycopene and its biosynthetic precursors [26]. These metabolites play critical roles in shaping the characteristic floral/fruity notes in tomato flavor profiles. In the present study, the S3 samples had the highest norisoprenoid content, with 6-methyl-5-hepten-2-one, geranylacetone, citral, and β-ionone identified as the most abundant norisoprenoid substances. The S3 samples had the highest concentrations of these volatiles and the S1 samples had the lowest. The differences in fruit aroma components likely reflect the fluctuations in light, temperature, or other environmental conditions during tomato plant growth. Notably, in a previous study, shading reduced norisoprenoid levels in Chardonnay grapes [44]. In another study, light exposure enhanced terpene and C13-norisoprenoid biosynthesis, but inhibited C6/C9 volatile production in grapes [17]. Leaf removal has been used to enhance sunlight exposure and stimulate norisoprenoids synthesis through promotion of carotenoid biosynthesis [45]. Similar results have been reported by Wang et al. [14] and Friedel et al. [15]. Significant associations were observed between the expression of *VvCCD1*, *VvCCD4a*, and *VvCCD4b* and the C13-norisoprenoids concentration in wine grapes [46]. Therefore, these genes/proteins are potentially modulated by light. Supporting this, research has demonstrated that light positively regulates carotenoid accumulation through transcriptional modulation of key metabolic genes in tomatoes and peppers [8], and strong light exposure enhances the activity of the *VvCCD4s* promoter [16]. However, a significant decrease in terpenoids was also observed in exposed berries, attributable to the elevated berry temperatures [10]. It should be noticed that norisoprenoids and their associated genes are regulated not only by light but also by temperature. Under low temperature storage, significant decreases in C13-norisporenoids (e.g., β-ionone) concentrations have been observed in raspberries [47]. Low-temperature conditions likely suppress the enzymatic activity of CCDs and reduce norisoprenoid biosynthesis. High temperatures increase the expression of CCD genes encoding carotenoid cleavage dioxygenases and thus enhance its enzymatic degradation, which results in a positive correlation with the levels of norisoprenoids [48]. However, another study found that transcriptional activation of the CCD4b promoter was suppressed under extremely high temperatures (>37 °C) [16]. Therefore, excessively high temperatures may lead to downregulation of gene expression and affect metabolite levels. Collectively, our results reflect the complex interplay between light and temperature in regulating norisoprenoid biosynthesis.

### 3.5. Principal Component Analysis

PCA was conducted to evaluate and categorize the volatile profiles of tomatoes with different cultivation periods, using all detected volatile compounds as variables. The first two principal components (PC1 and PC2) accounted for the majority of flavor characters of these three tomato samples (Figure 6). The Biplot illustrates the top 40 volatile substances (base on VIP values) that contribute most significantly to distinguishing the three samples. Hexanal, (*E*)-2-hexenal, nonanal, (*E*)-2-octenal, *trans*-geranylacetone, 6-methyl-5-hepten-2-one, 7-methyl-3,4-octadiene, and citral were found to be the key volatiles contributing most significantly to differentiating the samples across cultivation periods. These compounds imparted grassy and floral–fruity notes, respectively. In the PCA results, the S1 group was positioned on the negative axis of PC1, which demonstrated a positive correlation with higher contents of aldehydes, particularly hexanal. By contrast, the S2 and S3 groups were positioned along the positive direction of PC1, with further separation along PC2. S2 was located along the negative direction of the PC2 axis, and S3 was located along the positive direction of the PC2 axis. These results show that the S2 samples were characterized by fatty alcohols and some amino acid-derived volatile compounds (e.g., benzenoids such as benzaldehyde, phenylethyl alcohol). Meanwhile, the S3 samples were positively correlated with isoprenoid-derived volatiles—terpenoids and norisoprenoids.

## 4. Conclusions

A total of 227 volatile compounds, belonging to 9 aroma categories (grassy, fruity, floral, fatty, woody, nut-like, caramel, sour and others), were identified in tomatoes by HS-SPME Arrow-GC × GC-Q-TOF/MS. The dominant flavor profile in the tomatoes had vegetative/green and fatty characteristics, and this was complemented by secondary aromatic and floral notes. Among the compounds detected, hexanal and (*E*)-2-hexenal exhibited the highest relative concentrations and were assigned to fresh grass and green leaf aromas. This study reveals significant aroma variations among different cultivation periods. The S1 samples, which experienced the lowest temperature, solar radiation, and PAR compared with the other two samples, had a distinct grassy aroma but minimal floral/fruity notes. By contrast, the S3 samples had a significant floral and fruity fragrance. Most differential metabolites were linked to fatty acid, amino acid, and isoprenoid pathways. Lower temperature and light conditions upregulated fatty acid metabolism in the S1 samples. High temperatures and intense light promoted fatty alcohol accumulation in the S2 samples. The S3 samples were characterized by isoprenoid-derived volatiles, terpenoids, and norisoprenoids. These differences might be caused by fluctuations in the daily temperature and light intensity. Our findings suggest that light and temperature conditions can be optimized to enhance volatiles accumulation. Further investigations into the effects of light–temperature interactions on volatiles formation in tomatoes are required to improve our understanding of biosynthetic pathways and regulatory mechanisms during fruit ripening.

## Figures and Tables

**Figure 1 foods-14-02975-f001:**
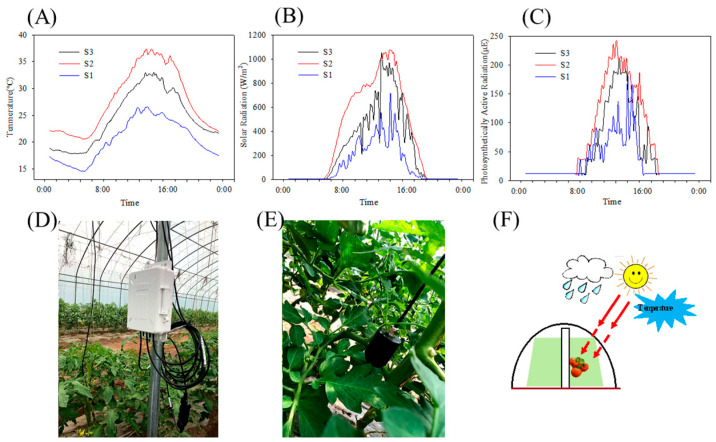
Changes in the temperature (**A**), solar radiation (**B**), and PAR (**C**). Meteorological data collectors (**D**,**E**) around the fruit cluster. (**F**) Simple diagram of greenhouse cultivation.

**Figure 2 foods-14-02975-f002:**
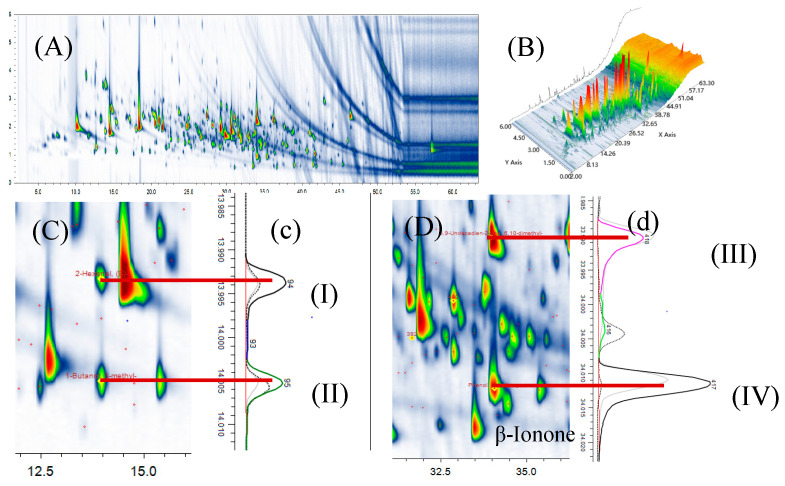
GC × GC-Q/TOFMS chromatograms of 2D and 3D diagram of volatiles from tomatoes (**A**,**B**); GC × GC plot (**C**,**D**) corresponding to the chromatography in the second dimension (**c**,**d**). I: 2-Hexenal, (E); II: 1-Butanol, 3-methyl-; III: 5,9-Undecadien-2-one,6,10-dimethyl-; IV: Phenol, 2-methoxy-.

**Figure 3 foods-14-02975-f003:**
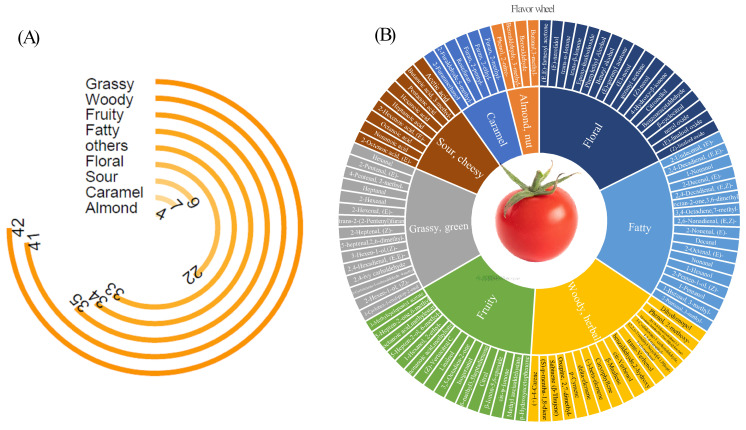
Analysis of volatiles in tomatoes. (**A**): radial bar chart representing the quantities of different aroma types; (**B**): the flavor wheel of key volatiles in tomatoes. Note: the outermost ring shows the names of the volatiles, and the middle ring shows the sensory flavor characteristics of volatile compounds.

**Figure 4 foods-14-02975-f004:**
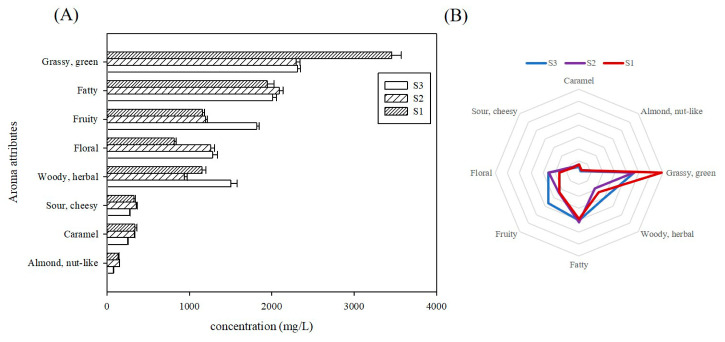
Variations in volatiles composition across different cultivation seasons. (**A**): Cumulative concentrations of distinct aroma classes; (**B**): radar map of aroma sensory characteristics of key volatiles in tomatoes. Note: the outermost ring indicates sensory flavor characteristics, and the broken line represents the cumulative contribution rate of semi-quantitative concentration of the corresponding type.

**Figure 5 foods-14-02975-f005:**
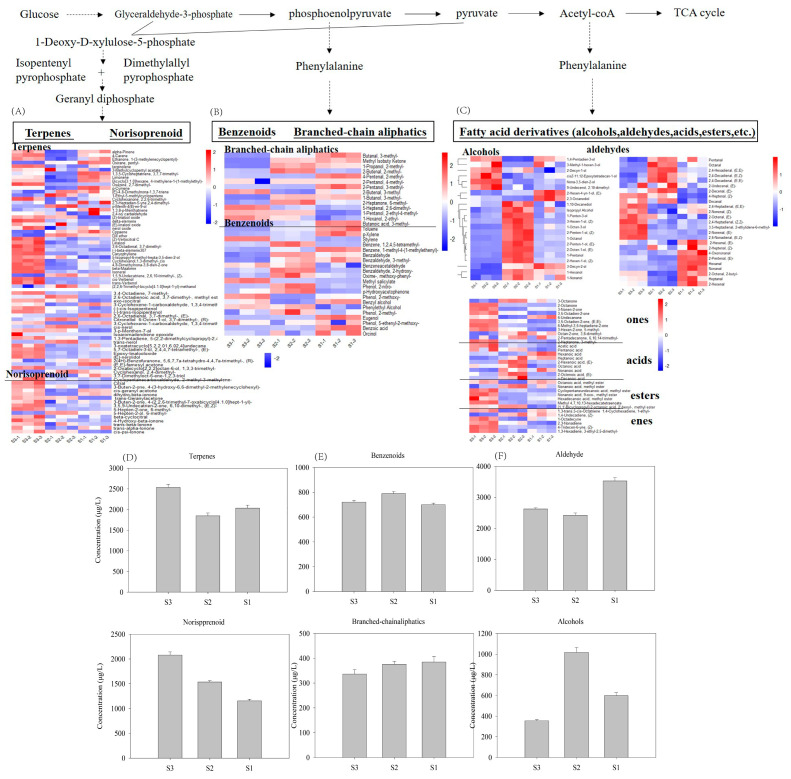
Volatile compounds in tomatoes among the three different cultivation times. (**A**–**C**) show the changes in isoprenoid-, amino acid-, and fatty acid-derived volatiles, respectively. (**D**–**F**) show the effect of harvest time on the total content of volatiles according to each metabolic pathway.

**Figure 6 foods-14-02975-f006:**
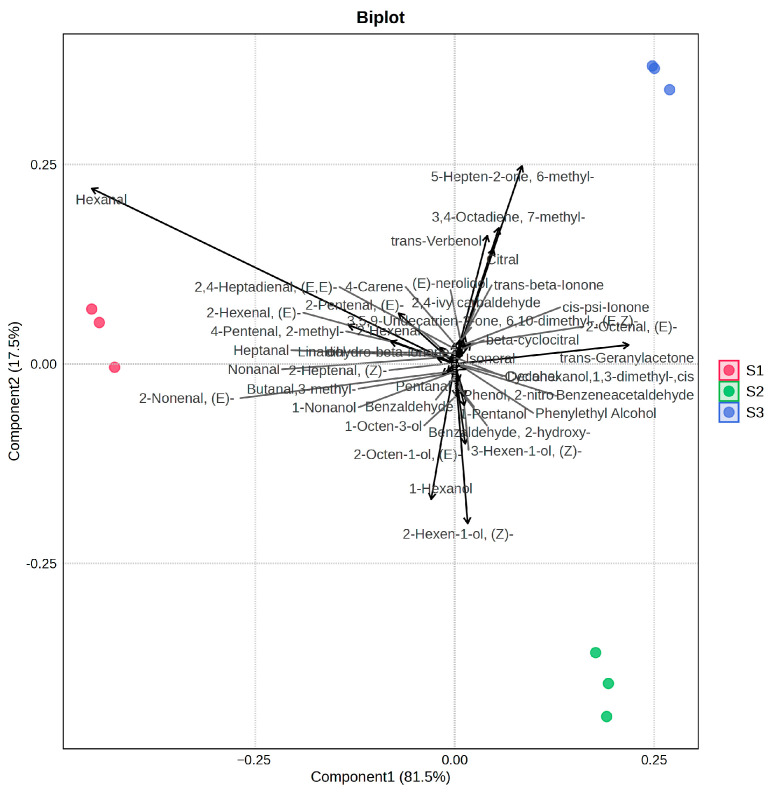
Principal component analysis of volatile compounds in tomatoes with different cultivation times.

## Data Availability

The original contributions presented in the study are included in the article/Appendix A, further inquiries can be directed to the corresponding author.

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
