# Peer review of "Flavor Profile of Tomatoes Across Different Cultivation Times Based on GC × GC-Q/TOFMS"

_foods, 2025, doi:10.3390/foods14172975_

Round 1
Reviewer 1 Report
Comments and Suggestions for Authors
I read the manuscript 'Flavor profile of tomatoes across different cultivation times based on comprehensive two-dimensional gas chromatography with quadrupole time-of-flight-mass spectrometry', investigated the impact of cultivation timing on tomato flavor profiles and also investigated to examining the regulatory roles of light and temperature within core metabolic pathways. The introduction is sufficient, well presents the importance of temperature and light management on tomato volatile profile.
In the title I would rather provide an abbreviation of the technique used, because it is not something new, it is already well recognized.
The abstract should at least generally explain how tomatoes, S1, S2 and S3 differ from each other. Also, please indicate what the aim of the research was.
Lines 53-56 - a bit of an illogical sentence because GCxGC is just two columns.
2.1 - please specify the tomato variety.
Line 133 - check the correct name of IS.
2.4 - there is no information whether the analyses were performed in splitless or split mode (and if so, its parameters).
3.1 - I don't think it's necessary to explain how two-dimensional gas chromatography works, it's a technique that's been known for over twenty years. Especially since the explanation provided is somewhat awkward, misleading.
3.2 please add the source from which the information about the aroma of individual compounds was taken.
3.3 "green-grassy odorants demonstrate the highest contribution" and other similar sentences. To present such conclusions, GC-O (olfactometry) must be performed or calculations based on concentrations and detection thresholds must be made. Without this, you can only indicate which compounds from a specific class were the most and which were the least, but you do not know what their share is in the smell perceived by consumers (aroma feature).
Lines 360-361 - Your research does not confirm this (because you did not examine the influence of other factors at the same time), nor do any of the sources you provided draw such conclusions. So I would be careful with giving such strong conclusions.
In studies that rely solely on GC-MS analyses, it is a good standard to provide a full table of identified compounds along with an indication of their experimental and literature retention Indices. This is important both in terms of checking the reliability of the analyses performed, but perhaps even more so - the convenience of using the data for other researchers to compare. Please add an appropriate table to the supplement.
Reviewer 2 Report
Comments and Suggestions for Authors
In this manuscript entitled “Flavor profile of tomatoes across different cultivation times based on comprehensive two-dimensional gas chromatography with quadrupole time-of-flight-mass spectrometry” authors have studied the variations in volatile components of three tomatoes samples cultivated at three different times and demonstrated the effects of cultivation periods on the volatiles of Tomatoes. The manuscript lacks serious issues with the characterization of the volatile components of studied tomatoes samples and must be carefully characterized all detected components providing the details of how each components were identified in Tabular format which must include retention time and LRIs values including experimental and the literature values.
Main concern of the manuscript which must be resolved before it processed further:
-Identification of the volatile components must be carried out in detail. For instance, authors should confirm the identification of each volatile components employing the LRIs values.
- Authors should carefully characterized all detected components providing the details of how each component were identified by providing a Table which must include retention time and LRIs values including experimental and the literature values, CAS no., Molecular formula and molecular mass and percentages of each components.
-Authors must calculate experimental LRI corresponding to each peak in GC-MS analysis using mixture of hydrocarbons.
-Authors must provide the GC chromatogram of each sample in 1D diagram indicating the peak of each identified compounds.
-Quantification of Volatile components should be compared based on the GC X GC analysis. For instance, if authors are get separation of a particular peak for example 1-Butanol, 3-methyl-(I) and 2-Hexenal (II) percentages must be detailed when they are separated from the second column. Same should be apply for all the components separated by second column.
-Chemical characterization parts of the manuscript must be improved.
-Most of the figures are blurred and not readable therefore quality of figures must be improved.
Comments on the Quality of English LanguageCould be improved
Round 2
Reviewer 1 Report
Comments and Suggestions for Authors
Judging by the table presented in the supplement, the chromatographic analyses and their data analysis were performed very carefully and thoroughly. I agree for accepting the manuscript. I suggest leaving only the second (cleaned) data sheet from the xlsx file for publication.
Author Response
Comments:Judging by the table presented in the supplement, the chromatographic analyses and their data analysis were performed very carefully and thoroughly. I agree for accepting the manuscript. I suggest leaving only the second (cleaned) data sheet from the xlsx file for publication.
Response: Thanks a lot for the suggestion. The first data sheet has been deleted from the supplementary Table S1. It was our oversight in retaining the raw data in the first sheet. Thank you again.
Thanks very much for giving us the opportunity to revise our manuscript, and we sincerely hope these revisions meet your requirements. We tried our best to improve the manuscript and proofread it by a native English speaker as well as a Language Editing Services provided by Liwen Bianji (Edanz) (www.liwenbianji.cn). And the certificate has been provided as supplementary file to submit. We hope the revised manuscript is satisfactory.
Reviewer 2 Report
Comments and Suggestions for Authors
Authors have improved the revised version of the manuscript and responded to most of the queries, however, there are still some correction required to improve the manuscript and they are given below,
-The authors should also add the content of the components in percentages in Table S1 for each samples to compare the results given in 1D chromatograms. It is difficult to quantify 227 components based on the standard compounds specially when there are so many components.
- verify total adds up to ~100% may be slightly less due to rounding or small undetected peaks.
-what is the unit of concentration of compounds for each samples given in TableS1.
-In Table S1 for LRI values from NIST Library authors have written “P” for many LRI values of compounds. What does it mean it should be explained in foot notes of the Table S1.
Author Response
Comments 1:- The authors should also add the content of the components in percentages in Table S1 for each samples to compare the results given in 1D chromatograms. It is difficult to quantify 227 components based on the standard compounds specially when there are so many components.
Response 1: The percentages of each component of each sample has been listed in Supplementary Table S1.
Comments 2:- verify total adds up to ~100% may be slightly less due to rounding or small undetected peaks.
Response 2: The total percentage adds up to less than 100% due to rounding and the removal of small undetected peaks, column loss peaks, and unidentified substances (lacking NIST RI data or matching scores < 750). A total of 986 peaks were detected, and 227 volatiles were identified in our study. The total percentage of original data (986 peaks) was 99.9% due to the rounding error. In addition, we excluded some undetected minor peaks, column loss peaks, and unidentifiable substances, thus resulting in a total percentage sum of less than 100%.
Comments 3:- what is the unit of concentration of compounds for each samples given in Table S1.
Response 3: The unit has been given in Table S1.
Comments 4:- In Table S1 for LRI values from NIST Library authors have written “P” for many LRI values of compounds. What does it mean it should be explained in foot notes of the Table S1.
Response 4: Letter“P”means the NIST RI comes from the standard polar column. It has been explained in foot notes of the Table S1.
Thanks very much for giving us the opportunity to revise our manuscript, and we sincerely hope these revisions meet your requirements. We tried our best to improve the manuscript and proofread it by a native English speaker as well as a Language Editing Services provided by Liwen Bianji (Edanz) (www.liwenbianji.cn). And the certificate has been provided as supplementary file to submit. We hope the revised manuscript is satisfactory.
